# Multi-agent Reinforcement Learning for Networked System Control

**Tianshu Chu**
Uhana Inc.
Palo Alto, CA 94304, USA
cts198859@hotmail.com

**Sandeep Chinchali & Sachin Katti**
Stanford University
Stanford, CA 94305, USA
{csandeep,skatti}@stanford.edu

## Abstract

This paper considers multi-agent reinforcement learning (MARL) in networked system control. Specifically, each agent learns a decentralized control policy based on local observations and messages from connected neighbors. We formulate such a networked MARL (NMARL) problem as a spatiotemporal Markov decision process and introduce a spatial discount factor to stabilize the training of each local agent. Further, we propose a new differentiable communication protocol, called NeurComm, to reduce information loss and non-stationarity in NMARL. Based on experiments in realistic NMARL scenarios of adaptive traffic signal control and cooperative adaptive cruise control, an appropriate spatial discount factor effectively enhances the learning curves of non-communicative MARL algorithms, while NeurComm outperforms existing communication protocols in both learning efficiency and control performance.

## 1 Introduction

Reinforcement learning (RL), formulated as a Markov decision process (MDP), is a promising data-driven approach for learning adaptive control policies (Sutton & Barto, 1998). Recent advances in deep neural networks (DNNs) further enhance its learning capacity on complex tasks. Successful algorithms include deep Q-network (DQN) (Mnih et al., 2015), deep deterministic policy gradient (DDPG) (Lillicrap et al., 2015), and advantage actor critic (A2C) (Mnih et al., 2016). However, RL is not scalable in many real-world control problems. This scalability issue is addressed in multi-agent RL (MARL), where each agent learns its individual policy from only local observations. However, MARL introduces new challenges in model training and execution, due to non-stationarity and partial observability in a decentralized MDP from the viewpoint of each agent. To address these challenges, various learning methods and communication protocols are proposed to stabilize training and improve observability.

This paper considers networked MARL (NMARL) in the context of networked system control (NSC), where agents are connected via a communication network for a cooperative control objective. Each agent performs decentralized control based on its local observations and messages from connected neighbors. NSC is extensively studied and widely applied. Examples include connected vehicle control (Jin & Orosz, 2014), traffic signal control (Chu et al., 2019), distributed sensing (Xu et al., 2018), and networked storage operation (Qin et al., 2016). We expect an increasing trend of NMARL based controllers in the near future, after the development of advanced communication technologies such as 5G and Internet-of-Things.

Recent works studied decentralized NMARL under assumptions of global observations and local rewards (Zhang et al., 2018; Qu et al., 2019), which are reasonable in multi-agent gaming but not suitable in NSC. First, the control infrastructures are distributed in a wide region, so collecting global observations in execution increases communication delay and failure rate, and hurts the robustness. Second, online learning is not common due to safety and efficiency concerns. Rather, each model is trained offline and tested extensively before field deployment. In online execution, the model only runs forward propagation, and its performance is constantly monitored for triggering re-training. To reflect these practical constraints in NSC, we assume 1) each agent is connected to a limited number

of neighbors and communication is restricted to its neighborhood, and 2) training is offline and global information is available in rollout training minibatches, despite a decentralized training process.

The contributions of this paper are three-fold. First, we formulate NMARL under the aforementioned NSC assumptions as a decentralized spatiotemporal MDP, and introduce a *spatial* discount factor to stabilize training, especially for non-communicative algorithms. Second, we propose a new neural communication protocol, called *NeurComm*, to adaptively share information on both system states and agent behaviors. Third, we design and simulate realistic NMARL environments to evaluate and compare our approaches against recent MARL baselines. [1]

## 2 RELATED WORK

MARL works can be classified into four groups based on their communication methods. The first group is non-communicative and focuses on stabilizing training with advanced value estimation methods. In MADDPG, each action-value is estimated by a centralized critic based on global observations and actions (or inferred actions) (Lowe et al., 2017). COMA extends the same idea to A2C and estimates each advantage using a centralized critic and a *counterfactual* baseline (Foerster et al., 2018). In Dec-HDRQN (Omidshafiei et al., 2017) and PS-TRPO (Gupta et al., 2017), the centralized critic takes local observations, but the parameters are shared globally. In the NMARL work of Zhang et al. (2018), the critic is fully decentralized but each takes global observations and performs *consensus* updates. In this paper, we empirically confirm that a spatial discount factor helps stabilize the training of non-communicative algorithms under neighborhood observation.

The second group considers heuristic communication protocols or direct information sharing. Foerster et al. (2017) shows performance gains with directly-shared low dimensional policy fingerprints from other agents. Similarly, mean field MARL takes the average of neighbor policies for informed action-value estimation (Yang et al., 2018). The major disadvantage of this group is that, unlike NeurComm, the communication is not explicitly designed for performance optimization, which may cause inefficient and redundant communications in execution.

The third group proposes learnable communication protocols. In DIAL, the message is generated together with action-value estimation by each DQN agent, then it is encoded and summed with other input signals at the receiver side (Foerster et al., 2016). CommNet is a more general communication protocol, but it calculates the mean of all messages instead of encoding them (Sukhbaatar et al., 2016). Both works, especially CommNet, incur an information loss due to aggregation on input signals. Another collection of works focuses on communications in strategy games. In BiCNet (Peng et al., 2017), a bi-directional RNN is used to enable flat communication among agents, while in Master-Slave (Kong et al., 2017), two-way message passing is utilized in a hierarchical RNN architecture of master and slave agents. In contrast to existing protocols, NeurComm 1) encodes and concatenates signals, instead of aggregating them, to minimize information loss, and 2) includes policy fingerprints in communication to reduce non-stationarity.

The fourth group focuses on communication *attentions* to selectively send messages. ATOC (Jiang & Lu, 2018) learns a soft attention which allocates a communication probability to each other agent, while IC3Net (Singh et al., 2018) learns a hard binary attention which decides communicating or not. These works are especially useful when each agent has to prioritize the communication targets. NMARL is less likely the case since the communication range is restricted to small neighborhoods.

## 3 SPATIOTEMPORAL RL

This section formulates the NMARL problem as a decentralized spatiotemporal MDP, and introduces the spatial discount factor to reduce its learning difficulty. To simplify the notation, we assume the true system state is observable, and use "state" and "observation" interchangeably. This does not affect the validity of proposed methods in practice. To save space, all proofs are deferred to A.

---

[1]Code link: `https://github.com/cts198859/deeprl_network`.

## 3.1 NETWORKED MARL

The networked system is represented by a graph $G(\mathcal{V}, \mathcal{E})$ where $i \in \mathcal{V}$ is each agent and $ij \in \mathcal{E}$ is each communication link. The corresponding MDP is characterized as $(G, \{\mathcal{S}_i, \mathcal{A}_i\}_{i \in \mathcal{V}}, p, r)$ where $\mathcal{S}_i$ and $\mathcal{A}_i$ are the local state space and action space of agent $i$. Let $\mathcal{S} := \times_{i \in \mathcal{V}} \mathcal{S}_i$ and $\mathcal{A} := \times_{i \in \mathcal{V}} \mathcal{A}_i$ be the global state space and action space, MDP transitions follow a stationary probability distribution $p : \mathcal{S} \times \mathcal{A} \times \mathcal{S} \to [0, 1]$, and global step rewards be denoted by $r : \mathcal{S} \times \mathcal{A} \to \mathbb{R}$. In a multi-agent MDP, each agent $i$ follows a decentralized policy $\pi_i : \mathcal{S}_i \times \mathcal{A}_i \to [0, 1]$ to chose its own action $a_{i,t} \sim \pi_i(\cdot|s_{i,t})$ at time $t$. The MDP objective is to maximize $\mathbb{E}[R_0^\pi]$, where $R_t^\pi = \sum_{\tau=t}^T \gamma^{\tau-t} r_\tau$ is the long-term global return with discount factor $\gamma$. Here the expectation is taken over the global policy $\pi : \mathcal{S} \times \mathcal{A} \to [0, 1]$, the initial distribution $s_t \sim \rho$, and the transition $s_{\tau+1} \sim p(\cdot|s_\tau, a_\tau)$, regarding the step reward $r_\tau = r(s_\tau, a_\tau), \forall \tau < T$, and the terminal reward $r_T = r_T(s_T)$ [2]. The same system can be formulated as a centralized MDP. Defining $V^\pi(s) = \mathbb{E}[R_t^\pi|s_t = s]$ as the *state-value* function and $Q^\pi(s, a) = \mathbb{E}[R_t^\pi|s_t = s, a_t = a]$ as the *action-value* function, we have $\mathbb{E}[R_0^\pi] = \sum_{s \in \mathcal{S}} \rho(s) V^\pi(s)$, $V^\pi(s) = \sum_{a \in \mathcal{A}} \pi(a|s) Q^\pi(s, a)$, and the *advantage* function $A^\pi(s, a) = Q^\pi(s, a) - V^\pi(s)$.

MARL provides a scalable solution for controlling networked systems, but it introduces partial observability and non-stationarity in decentralized MDP of each agent, leading to inefficient and unstable learning performance. To see this, note $s_{i,t} \in \mathcal{S}_i \subseteq \mathcal{S}$ does not provide sufficient information for $\pi_i$. Even assuming $s_{i,t} = s_t$, the transition $p_i(s_{i,t+1}|s_{i,t}, a_{i,t}) = \sum_{a_{-i,t} \in \mathcal{A}_{-i}} \pi_{-i}(a_{-i,t}|s_t) \cdot p(s_{t+1}|s_t, a_{i,t}, a_{-i,t})$ is non-stationary if the behavior policies of other agents $\pi_{-i} := \{\pi_j\}_{j \in \mathcal{V} \setminus \{i\}}$ are evolving over time. In this paper, we enforce practical constraints and only allow local observations and neighborhood communications, which makes MARL even more challenging.

**Definition 3.1** (Networked Multi-agent MDP with Neighborhood Communication). In a networked cooperative multi-agent MDP $(G, \{\mathcal{S}_i, \mathcal{A}_i\}_{i \in \mathcal{V}}, \{\mathcal{M}_{ij}\}_{ij \in \mathcal{E}}, p, \{r_i\}_{i \in \mathcal{V}})$ with the message space $\mathcal{M}$, the global reward is defined as $r = \frac{1}{|\mathcal{V}|} \sum_{i \in \mathcal{V}} r_i$. All local rewards are shared globally, whereas the communication is limited to neighborhoods, that is, each agent $i$ observes $\tilde{s}_{i,t} := s_{i,t} \cup m_{\mathcal{N}_i i,t}$. Here $\mathcal{N}_i := \{j \in \mathcal{V}|ji \in \mathcal{E}\}$, $m_{\mathcal{N}_i i,t} := \{m_{ji,t}\}_{j \in \mathcal{N}_i}$, and each message $m_{ji,t} \in \mathcal{M}_{ji}$ is derived from all the available information at that neighbor.

## 3.2 SPATIOTEMPORAL RL

**Definition 3.2** (Spatiotemporal MDP). We assume local transitions are independent of other agents given the neighboring agents, that is,

$$p_i(s_{i,t+1}|s_{\mathcal{V}_i,t}, a_{i,t}) = \sum_{a_{\mathcal{N}_i,t} \in \mathcal{A}_{\mathcal{N}_i}} \prod_{j \in \mathcal{N}_i} \pi_j(a_{j,t}|\tilde{s}_{j,t}) \cdot p(s_{i,t+1}|s_{\mathcal{V}_i,t}, a_{i,t}, a_{\mathcal{N}_i,t}), \tag{1}$$

where $\mathcal{V}_i := \mathcal{N}_i \cup \{i\}$ is the closed neighborhood, and $p$ is abused to denote any stationary transition. Then from the viewpoint of each agent $i$, Definition 3.1 is equivalent to a decentralized spatiotemporal MDP, characterized as $(\mathcal{S}_i, \mathcal{A}_i, \{\mathcal{M}_{ji}\}_{j \in \mathcal{N}_i}, p_i, \{r_i\}_{i \in \mathcal{V}})$, by optimizing the discounted return

$$R_{i,t}^\pi = \sum_{\tau=t}^T \gamma^{\tau-t} \left( \sum_{j \in \mathcal{V}} \alpha^{d_{ij}} r_{j,t} \right), \tag{2}$$

where $0 \leq \alpha \leq 1$ is the spatial discount factor, and $d_{ij}$ is distance between agents $i$ and $j$.

The major assumption in Definition 3.2 is that the Markovian property holds both temporally and spatially, so that the next local state depends on the neighborhood states and policies only. This assumption is valid in most networked control systems such as traffic and wireless networks, as well as the power grid, where the impact of each agent is spread over the entire system via controlled flows, or chained local transitions. Note in NSC, each agent is connected to a limited number of neighbors (the degree of $G$ is low). So spatiotemporal MDP is decentralized during model execution, and it naturally extends properties of MDP. To reduce the learning difficulty of spatiotemporal MDP, a spatiotemporally discounted return is introduced in Eq. (2) to scale down reward signals further away (which are more difficult to fit using local information). When $\alpha \to 0$,

---

[2]In infinite MDP, $r_T(s) = \mathbb{E}\left[\sum_{t=T}^\infty \gamma^{t-T} r_t \Big| s_T = s\right]$.

each agent performs local greedy control; when $\alpha \to 1$, each agent performs global coordination and $R_{i,t}^\pi = R_t^\pi, \forall i \in \mathcal{V}$. Further, we have $Q_i^\pi(s,a) = Q_i^\pi(s, a_{\mathcal{V}_i}) = \mathbb{E}[R_{i,t}^\pi | s_t = s, a_{\mathcal{V}_i,t} = a_{\mathcal{V}_i}]$, and $V_i^\pi(s, a_{-i}) = V_i^\pi(s, a_{\mathcal{N}_i}) = \sum_{a_i \in \mathcal{A}_i} \pi_i(a_i | \tilde{s}_i) Q_i^\pi(s, a_{\mathcal{V}_i})$, since the immediate local reward of each agent is only affected by controls within its closed neighborhood.

Now we assume each agent is A2C, with parametric models $\pi_{\theta_i}(\tilde{s}_i)$ and $V_{\omega_i}(\tilde{s}_i, a_{\mathcal{N}_i})$ for fitting the optimal policy $\pi_i^*$ and value function $V^{\pi_i}$. Note if $\tilde{s}_i$ is able to provide global information through cascaded neighborhood communications, both $\pi_{\theta_i}$ and $V_{\omega_i}$ are able to fit return $R_{i,t}^\pi$. Also, global and future information, such as $R_{i,\tau}^\pi$ and $a_{\mathcal{N}_i,\tau}$, are always available from each rollout minibatch in offline training. In contrast, only local information $\tilde{s}_{i,t}$ is allowed in online execution of policy $\pi_{\theta_i}$.

**Proposition 3.1** (Spatiotemporal RL with A2C). Let $\{\pi_{\theta_i}\}_{i \in \mathcal{V}}$ and $\{V_{\omega_i}\}_{i \in \mathcal{V}}$ be the decentralized actor-critics, and $\{(s_{i,\tau}, m_{\mathcal{N}_i i,\tau}, a_{i,\tau}, r_{i,\tau})\}_{i \in \mathcal{V}, \tau \in \mathcal{B}}$ be the on-policy minibatch from spatiotemporal MDPs under stationary policies $\{\pi_{\theta_i}\}_{i \in \mathcal{V}}$. Then each actor and critic are updated by losses

$$\mathcal{L}(\theta_i) = \frac{1}{|\mathcal{B}|} \sum_{\tau \in \mathcal{B}} \left( -\log \pi_{\theta_i}(a_{i,\tau}|\tilde{s}_{i,\tau}) \hat{A}_{i,\tau}^\pi + \beta \sum_{a_i \in \mathcal{A}_i} \pi_{\theta_i}(a_i|\tilde{s}_{i,\tau}) \log \pi_{\theta_i}(a_i|\tilde{s}_{i,\tau}) \right), \quad (3)$$

$$\mathcal{L}(\omega_i) = \frac{1}{|\mathcal{B}|} \sum_{\tau \in \mathcal{B}} \left( \hat{R}_{i,\tau}^\pi - V_{\omega_i}(\tilde{s}_{i,\tau}, a_{\mathcal{N}_i,\tau}) \right)^2, \quad (4)$$

where $\hat{A}_{i,\tau}^\pi = \hat{R}_{i,\tau}^\pi - v_{i,\tau}$ is the estimated advantage, $\hat{R}_{i,\tau}^\pi = \sum_{\tau'=\tau}^{\tau_\mathcal{B}-1} \gamma^{\tau'-\tau} \left( \sum_{j \in \mathcal{V}} \alpha^{d_{ij}} r_{j,\tau'} \right) + \gamma^{\tau_\mathcal{B}-\tau} v_{i,\tau_\mathcal{B}}$ is the sampled action-value, $v_{i,\tau} = V_{\omega_i^-}(\tilde{s}_{i,\tau}, a_{\mathcal{N}_i,\tau})$ is the estimated state-value, and $\beta$ is the coefficient of the entropy loss.

# 4 SPATIOTEMPORAL RL WITH NEURAL COMMUNICATION

For efficient and adaptive information sharing, we propose a new communication protocol called NeurComm. To simplify the notation, we assume all messages sent from agent $i$ are identical, *i.e.*, $m_{ij} = m_i, \forall j \in \mathcal{N}_i$. Then

$$h_{i,t} = g_{\nu_i}(h_{i,t-1}, e_{\lambda_i^s}(s_{\mathcal{V}_i,t}), e_{\lambda_i^p}(\pi_{\mathcal{N}_i,t-1}), e_{\lambda_i^h}(h_{\mathcal{N}_i,t-1})), \quad (5)$$

where $h_{i,t}$ is the hidden state (or the *belief*) of each agent and $e_{\lambda_i}$ and $g_{\nu_i}$ are differentiable message encoding and extracting functions [3]. To avoid dilution of state and policy information (the former is for improving observability while the later is for reducing non-stationarity), state and policy are explicitly included in the message besides agent belief, *i.e.*, $m_{i,t} = s_{i,t} \cup \pi_{i,t-1} \cup h_{i,t-1}$, or $\tilde{s}_{i,t} := s_{\mathcal{V}_i,t} \cup \pi_{\mathcal{N}_i,t-1} \cup h_{\mathcal{N}_i,t-1}$ as in Eq. (5). Note the communication phase is prior-decision, so only $h_{i,t-1}$ and $\pi_{i,t-1}$ are available. This protocol can be easily extended for multi-pass communication: $h_{i,t}^{(k)} = g_{\nu_i^{(k)}}(h_{i,t}^{(k-1)}, e_{\lambda_i^s}(s_{\mathcal{V}_i,t}), e_{\lambda_i^p}(\pi_{\mathcal{N}_i,t-1}), e_{\lambda_i^h}(h_{\mathcal{N}_i,t}^{(k-1)}))$, where $h_{i,t}^{(0)} = h_{i,t-1}$, and $k$ denotes each of the communication passes. The communication attentions can be integrated either at the sender as $\mu_{i,t}(m_{i,t})$, or at the receiver as $\mu_{i,t}(m_{\mathcal{N}_i,t})$. Replacing the input ($\tilde{s}_{i,t}$) of Eq. (3)(4) with the belief ($h_{i,t}$), the actor and critic become $\pi_{\theta_i}(\cdot|h_{i,t})$ and $V_{\omega_i}(h_{i,t}, a_{\mathcal{N}_i,t})$, and the frozen estimations are $\pi_{i,t}$ and $v_{i,t}$, respectively.

**Proposition 4.1** (Neighborhood Neural Communication). In spatiotemporal RL with neighborhood NeurComm, each agent utilizes the delayed global information to learn its belief, and it learns the message to optimize the control performance of all other agents.

NeurComm enabled MARL can be represented using a single meta-DNN since all agents are connected by differentiable communication links, and $\tilde{s}_i$ are the intermediate outputs after communication layers. Fig. 1a illustrates the forward propagations inside each individual agent and Fig. 1b shows the broader multi-step spatiotemporal propagations. Note the gradient propagation of this meta-DNN is decentralized based on each local loss signal. As time advances, the involved parameters in each propagation expand spatially in the meta-DNN, due to the cascaded neighborhood communication. To see this mathematically, $\pi_{\theta_{i,t}}(\cdot|h_{i,t}) = \pi_{\tilde{\theta}_{i,t}}(\cdot|s_{\mathcal{V}_i,t}, \pi_{\mathcal{N}_i,t-1})$, with $\tilde{\theta}_{i,t} = \{\lambda_i, \nu_i, \theta_i\}$; while $\pi_{\theta_{i,t+1}}(\cdot|h_{i,t+1}) = \pi_{\tilde{\theta}_{i,t+1}}(\cdot|s_{\mathcal{V}_i,t+1}, \pi_{\mathcal{N}_i,t}, \{s_{\mathcal{N}_j,t}, \pi_{\mathcal{N}_j,t-1}\}_{j \in \mathcal{N}_i})$, with

---
[3]Additional cell state needs to be maintained if LSTM is used.

$\tilde{\theta}_{i,t+1} = \{\lambda_j, \nu_j\}_{j \in \mathcal{N}_i} \cup \{\lambda_i, \nu_i, \theta_i\}$. In other words, $\{\lambda_i, \nu_i\}$ will be updated for improving actors $\pi_{\theta_j}, \forall j \in \mathcal{V}$, as soon as they are included in $\tilde{\theta}_j$; meanwhile, $r_i$ will be included in $R_j^\pi$. In contrast, the policy is fully decentralized in execution, as $g_{\nu_i}$ depends on $\tilde{s}_i$ only.

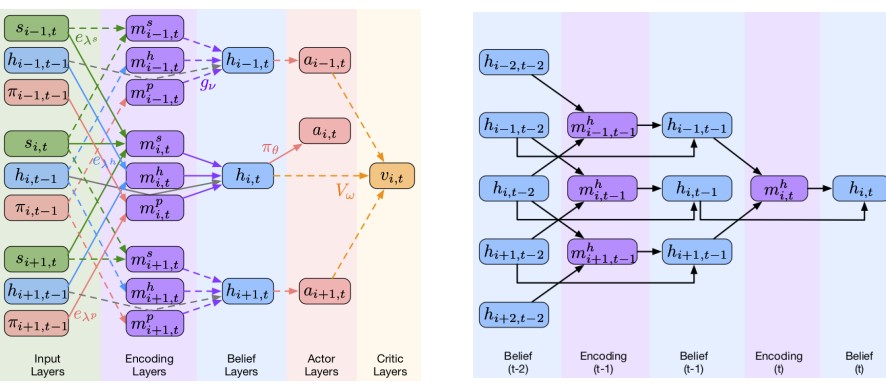

(a) Intra-step propagations.            (b) Inter-step propagations.

Figure 1: Forward propagations of NeurComm enabled MARL, illustrated in a queueing system. (a) Single-step forward propagations inside agent $i$. Different colored boxes and arrows show different outputs and functions, respectively. Solid and dashed arrows indicate actor and critic propagations, respectively. (b) Multi-step forward propagations for updating the belief of agent $i$.

NeurComm is general enough and has connections to other communication protocols. CommNet performs a more lossy aggregation since the received messages are averaged before encoding, and all encoded inputs are summed up (Sukhbaatar et al., 2016). In DIAL, each DQN agent encodes the received messages instead of averaging them, but still it sums all encoded inputs (Foerster et al., 2016). Also, both CommNet and DIAL do not have policy fingerprints included in messages.

## 5 NUMERICAL EXPERIMENTS

### 5.1 ENVIRONMENT SETUP

There are several benchmark MARL environments such as cooperative navigation and predator-prey, but few of them represent NSC. Here we design two NSC environments: adaptive traffic signal control (ATSC) and cooperative adaptive cruise control (CACC). Both ATSC and CACC are extensively studied in intelligent transportation systems, and they hold assumptions of a spatiotemporal MDP.

### 5.1.1 ADAPTIVE TRAFFIC SIGNAL CONTROL

The objective of ATSC is to adaptively adjust signal phases to minimize traffic congestion based on real-time road-traffic measurements. Here we implement two ATSC scenarios: a $5 \times 5$ synthetic traffic grid and a real-world 28-intersection traffic network from Monaco city, using standard microscopic traffic simulator SUMO (Krajzewicz et al., 2012).

**General settings.** For both scenarios, each episode simulates the peak-hour traffic, and a 5s control interval is applied to prevent traffic light from too frequent switches, based on RL control latency and driver response delay. Thus, one MDP step corresponds to 5s simulation and the horizon is 720 steps. Further, a 2s yellow time is inserted before switching to red light for safety purposes. In ATSC, the real-time traffic flow, that is, the total number of approaching vehicles along each incoming lane, is measured by near-intersection induction-loop detectors (ILDs) (shown as the blue areas of example intersections in Fig. 2). The cost of each agent is the sum of queue lengths along all incoming lanes.

**Scenario settings.** Fig. 2a illustrates the traffic grid formed by two-lane arterial streets with speed limit 20m/s and one-lane avenues with speed limit 11m/s. We simulate the peak-hour traffic dynamics through four collections of time-variant traffic flows, with both loading and recovering phases. At beginning, three major flows $F_1$ are generated with *origin-destination* (O-D) pairs $x_{10}$-$x_4$, $x_{11}$-$x_5$, and $x_{12}$-$x_6$, meanwhile three minor flows $f_1$ are generated with O-D pairs $x_1$-$x_7$, $x_2$-$x_8$, and $x_3$-$x_9$.

After 15 minutes, $F_1$ and $f_1$ start to decay, while their opposite flows $F_2$ and $f_2$ start to dominate, as shown in Fig. 2b. Note the flows define the high-level demand only, the particular route of each vehicle is randomly generated. The grid is homogeneous and all agents have the same action space, which is a set of five pre-defined signal phases. Fig. 2c illustrates the Monaco traffic network, with controlled intersections in blue. NMARL in this scenario is more challenging since the network is heterogeneous with a variety of observation and action spaces. Four traffic flow collections are generated to simulate the peak-hour traffic, and each flow is a multiple of a "unit" flow of 325veh/hr, with randomly sampled O-D pairs inside rectangle areas in Fig. 2c. $F_1$ and $F_2$ are simulated during the first 40min, as $[1, 2, 4, 4, 4, 4, 2, 1]$ unit flows with 5min intervals; $F_3$ and $F_4$ are generated in the same way, but with a delay of 15min. See code for more details.

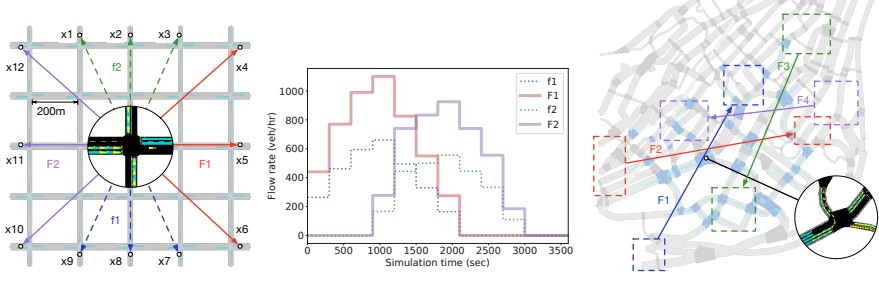

(a) Synthetic traffic grid. (b) Traffic flows within the grid. (c) Monaco traffic network.

Figure 2: ATSC scenarios for NMARL. (a) Synthetic traffic grid, with major and minor traffic flows shown in solid and dotted arrows. (b) Simulated time-variant traffic flows within the traffic grid. (c) Monaco traffic network, with traffic flow collections shown in colored arrows.

### 5.1.2 Cooperative Adaptive Cruise Control

The objective of CACC is to adaptively coordinate a platoon of vehicles to minimize the car-following headway and speed perturbations based on real-time vehicle-to-vehicle communication. Here we implement two CACC scenarios: "Catch-up" and "Slow-down", with physical vehicle dynamics.

**General settings.** For both CACC tasks, we simulate a string of 8 vehicles for 60s, with a 0.1s control interval. Each vehicle observes and shares its headway h, velocity v, and acceleration a to neighbors within two steps. The safety constraints are: $h \geq 1m$, $v \leq 30m/s$, $|a| \leq 2.5m/s^2$. Safe RL is relevant here, but itself is a big topic and out of the scope of this paper. So we adopt a simple heuristic *optimal velocity model* (OVM) (Bando et al., 1995) to perform longitudinal vehicle control under above constraints, whose behavior is affected by hyper-parameters: headway gain $\alpha^\circ$, relative velocity gain $\beta^\circ$, stop headway $h_{st} = 5m$ and full-speed headway $h_{go} = 35m$. Usually $(\alpha^\circ, \beta^\circ)$ represent the human driver behavior, here we train NMARL to recommend appropriate $(\alpha^\circ, \beta^\circ)$ for each OVM controller, selected from four levels $\{(0,0), (0.5,0), (0,0.5), (0.5,0.5)\}$. Assuming the target headway and velocity profile are $h^* = 20m$ and $v_t^*$, respectively, the cost of each agent is $(h_{i,t} - h^*)^2 + (v_{i,t} - v_t^*)^2 + 0.1u_{i,t}^2$. Whenever a collision happens ($h_{i,t} < 1m$), a large penalty of 1000 is assigned to each agent and the state becomes absorbing. An additional cost $5(2h_{st} - h_{i,t})_+^2$ is provided in training for potential collisions.

**Scenario settings.** Since exploring a collision-free CACC strategy itself is challenging for on-policy RL, we consider simple scenarios. In Catch-up scenario, $v_{i,0} = v_t^* = 15m/s$ and $h_{i,0} = h^*$, $\forall i \neq 1$, whereas $h_{1,0} = a \cdot h^*$, with $a \in U[3, 4]$. In Slow-down scenario, $v_{i,0} = v_0^* = b \cdot 15m/s$, $b \in U[1.5, 2.5]$, and $h_{i,0} = h^*$, $\forall i$, whereas $v_t^*$ linearly decreases to 15m/s during the first 30s and then stays at constant.

### 5.2 Algorithm Setup

For fair comparison, all MARL approaches are applied to A2C agents with learning methods in Eq. (3)(4), and only neighborhood observation and communication are allowed. **IA2C** performs independent learning, which is an A2C implementation of MADDPG (Lowe et al., 2017) as the critic takes neighboring actions (see Eq. (4)). **ConseNet** (Zhang et al., 2018) has the additional consensus

update to overwrite parameters of each critic as the mean of those of all critics inside the closed neighborhood. **FPrint** (Foerster et al., 2017) includes neighbor policies. **DIAL** (Foerster et al., 2016) and **CommNet** (Sukhbaatar et al., 2016) are described in Section 4. IA2C, ConseNet, and FPrint are non-communicative policies since they utilize only neighborhood information. In contrast, DIAL, CommNet, and NeurComm are communicative policies. Note communicative policies require more messages to be transferred and so higher communication bandwidth. In particular, the local message sizes are $O(|s_i| + |\pi_i| + |h_i|)$ for DIAL and NeurComm, $O(|s_i| + |h_i|)$ for CommNet, $O(|s_i| + |\pi_i|)$ for FPrint, and $O(|s_i|)$ for IA2C and ConseNet. The implementation details are in C.1.

All algorithms use the same DNN hidden layers: one fully-connected layer for message encoding $e_\lambda$, and one LSTM layer for message extracting $g_\nu$. All hidden layers have 64 units. The encoding layer implicitly learns normalization across different input signal types. We train each model over 1M steps, with $\gamma = 0.99$, actor learning rate $5 \times 10^{-4}$, and critic learning rate $2.5 \times 10^{-4}$. Also, each training episode has a different seed for generalization purposes. In ATSC, $\beta = 0.01$, $|\mathcal{B}| = 120$, while in CACC, $\beta = 0.05$, $|\mathcal{B}| = 60$, to encourage the exploration of collision-free policies. Each training takes about 30 hours on a 32GB memory, Intel Xeon CPU machine.

## 5.3 Ablation Study

We perform ablation study in proposed scenarios, which are sorted as ATSC Monaco > ATSC Grid > CACC Slow-down > CACC Catch-up by task difficulty. ATSC is more challenging than CACC due to larger scale (>=25 vs 8), more complex dynamics (stochastic traffic flow vs deterministic vehicle dynamics), and longer control interval (5s vs 0.1s). ATSC Monaco > ATSC Grid due to more heterogenous network, while CACC Slow-down > CACC Catch-up due to more frequently changing leading vehicle profile. To visualize the learning performance, we plot the learning curve, that is, average episode return ($\bar{R} = \frac{1}{T} \sum_{t=0}^{T-1} \sum_{i \in \mathcal{V}} r_{i,t}$) vs training step. For better visualization, all learning curves are smoothened using moving average with a window size of 100 episodes.

First, we investigate the impact of spatial discount factor, by comparing the learning curves among $\alpha \in \{0.8, 0.9, 1\}$ for IA2C and CommNet. Fig. 3 reveals a few interesting facts. First, $\alpha^*_{\text{CommNet}}$ is always higher than $\alpha^*_{\text{IA2C}}$. Indeed, $\alpha^*_{\text{CommNet}} = 1$ in almost all scenarios (except for ATSC Monaco). This is because communicative policies perform delayed global information sharing, whereas non-communicative policies utilize neighborhood information only, causing difficulty to fit the global return. Second, learning performance becomes much more sensitive to $\alpha$ when the task is more difficult. Specifically, all $\alpha$ values lead to similar learning curves in CACC Catch-up, whereas appropriate $\alpha$ values help IA2C converge to much better policies more steadily in other scenarios. Third, $\alpha^*$ is high enough: $\alpha^*_{\text{IA2C}} = 0.9$ except for CACC Slow-down where $\alpha^*_{\text{IA2C}} = 0.8$. This is because the discounted problem must be similar enough to the original problem in execution.

Next, we investigate the impact of NeurComm under $\alpha = 1$. We start with a baseline which is similar to existing differentiable protocols, i.e., $h_{i,t} = \texttt{LSTM}(h_{i,t-1}, \texttt{relu}(s_{\mathcal{V}_i,t}) + \texttt{relu}(m_{\mathcal{N}_i,t}))$. We then evaluate two intermediate protocols "Concat Only" and "FPrint Only", in which encoded inputs are concatenated and neighbor policies are included, respectively. Finally we evaluate their combination NeurComm. As shown in Fig. 3, all protocols have similar learning curves in easy CACC Catch-up scenario. Otherwise, both "Concat" and "FPrint" are able to enhance the baseline learning curves in certain scenarios and their affects are additive in NeurComm.

## 5.4 Training Results

Fig. 4 compares the learning curves of all MARL algorithms, after tuned $\alpha^* \in \{0.6, 0.8, 0.9, 0.95, 1\}$. As expected, $\alpha^*$ for non-communicative policies are lower than those for communicative policies.

Table 1: Best spatial discount factors $\alpha^*$ across NMARL scenarios.

| Scenario Name | NeurComm | CommNet | DIAL | IA2C | FPrint | ConseNet |
|---|---|---|---|---|---|---|
| ATSC Grid | 1.0 | 1.0 | 1.0 | 0.9 | 0.95 | 0.9 |
| ATSC Monaco | 1.0 | 0.9 | 0.9 | 0.9 | 0.9 | 0.9 |
| CACC Catch-up | 1.0 | 1.0 | 1.0 | 1.0 | 1.0 | 1.0 |
| CACC Slow-down | 1.0 | 1.0 | 1.0 | 0.8 | 0.9 | 0.8 |

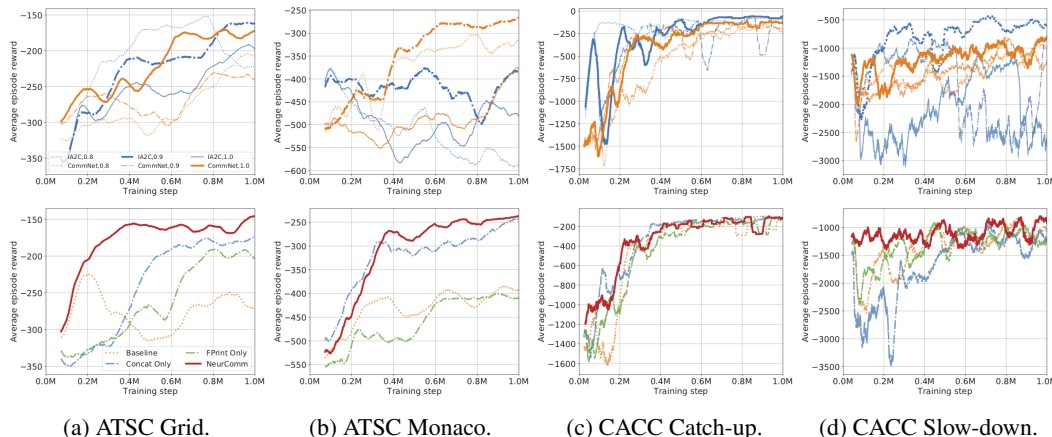

Figure 3: Sensitivity and ablation study of spatial discount factor (top) and NeurComm (bottom). The best learning curves are in bold.

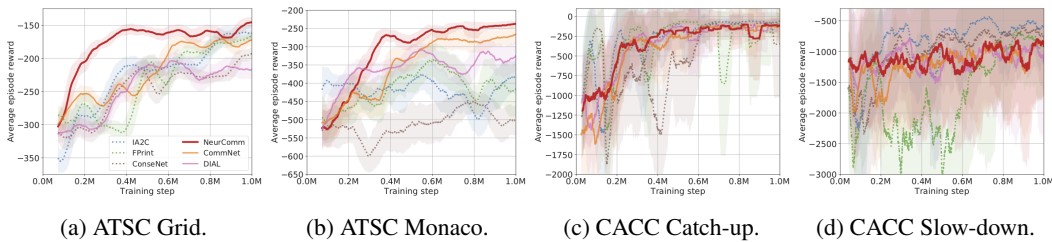

Figure 4: Training performance comparison after tuned spatial discount factors.

Tab. 1 summarizes $\alpha^*$ of controllers across different NMARL scenarios. For challenging scenarios like ATSC Monaco, lower $\alpha$ is preferred by almost all policies (except NeurComm). This demonstrates that $\alpha$ is an effective way to enhance MARL performance in general, especially for challenging tasks like ATSC Monaco. From another view point, $\alpha$ serves as an informative indicator on problem difficulty and algorithm coordination level. Based on Fig. 4, NeurComm is at least competitive in CACC scenarios, and it clearly outperforms other policies on both sample efficiency and learning stability in more challenging ATSC scenarios. Note in CACC a big penalty is assigned whenever a collision happens, so the standard deviation of episode returns is high.

## 5.5 EXECUTION RESULTS

We freeze and evaluate trained MARL policies in another 50 episodes, and summarize the results in Tab. 2. In CACC scenarios, $\alpha$ enhanced FPrint policy achieves the best execution performance. Note NeurComm still outperforms other communicative algorithms, so this result implies that delayed information sharing may not be helpful in easy but real-time and safety-critical CACC tasks. In contrast, NeurComm achieves the best execution performance for ATSC tasks. We also evaluate the execution performance of ATSC and CACC using domain-specific metrics in Tab. 3 and Tab. 4, respectively. The results are consistent with the reward-defined ones in Tab. 2.

Further, we investigate the performance of top policies in ATSC scenarios. For each ATSC scenario, we select the top two non-communicative and communicative policies and visualize their impact on network traffic by plotting the time series of network averaged queue length and intersection delay in Fig. 5. Note the line and shade show the mean and standard deviation of each metric across execution runs, respectively. Based on Fig. 5a, NeurComm achieves the most sustainable traffic control in ATSC Grid, so that the congested grid starts recovering immediately after the loading phase ends at 3000s. During the same unloading phase, CommNet prevents the queues from further increasing while non-communicative policies are failed to do so. Also, FPrint is less robust than IA2C as it

Table 2: Execution performance comparison over trained MARL policies. Best values are in bold.

| Scenario Name | NeurComm | CommNet | DIAL | IA2C | FPrint | ConseNet |
|---|---|---|---|---|---|---|
| ATSC Grid | **-136.1** | -165.1 | -214.4 | -160.2 | -155.9 | -187.5 |
| ATSC Monaco | **-226.3** | -263.0 | -339.4 | -369.7 | -359.4 | -528.9 |
| CACC Catch-up | -94.6 | -95.6 | -246.4 | -261.7 | **-57.8** | -419.7 |
| CACC Slow-down | -934.7 | -950.8 | -1112 | -2209 | **-697.9** | -1038 |

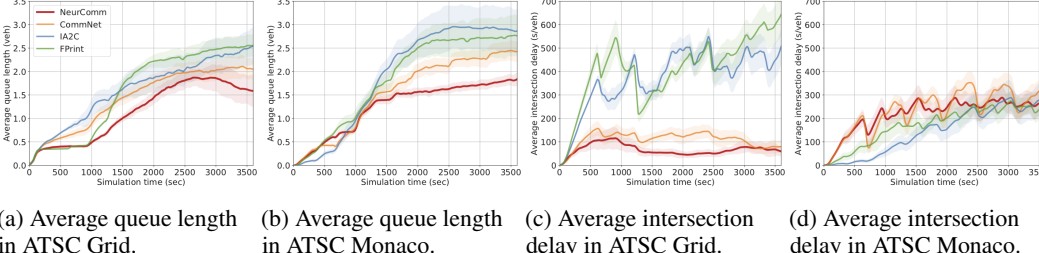

(a) Average queue length in ATSC Grid.

(b) Average queue length in ATSC Monaco.

(c) Average intersection delay in ATSC Grid.

(d) Average intersection delay in ATSC Monaco.

Figure 5: Execution performance comparison among top policies in ATSC scenarios, measured as average queue length and average intersection delay over time.

introduces a sudden congestion jump at 1000s. Similarly, NeurComm achieves the lowest saturation rate in ATSC Monaco (Fig. 5b).

Intersection delay is another key metric in ATSC. Based on Fig. 5c, communicative policies are able to reduce intersection delay as well in ATSC Grid, though it is not explicitly included in the objective and so is not optimized by non-communicative policies. In contrast, communicative policies have fast increase on intersection delay in ATSC Monaco. This implies that communicative algorithms are able to capture the spatiotemporal traffic pattern in homogeneous networks whereas they still have the risk of overfitting on queue reduction in realistic and heterogenous networks. For example, they block the short source edges on purpose to reduce on-road vehicles by paying a small cost of queue length.

Finally, we investigate the robustness (string stability) of top policies in CACC scenarios. In particular, we plot the time series of headway and velocity for the first and the last vehicles in the platoon. The profile of the first vehicle indicates how adaptively the controller pursues $h^*$ and $v^*$, while that of the last vehicle indicates how stable the controlled platoon is. Based on Tab. 1 and Tab. 4, the top communicative and non-communicative controllers are NeurComm and FPrint.

Fig. 6 shows the corresponding headway and velocity profiles for the selected controllers. Interestingly, MARL controllers are able to achieve steady state $v^*$ and $h^*$ for the first vehicle of platoon, whereas they still have difficulty to eliminate the perturbation through the platoon. This may be because of the heuristic low-level controller as well as the delayed information sharing.

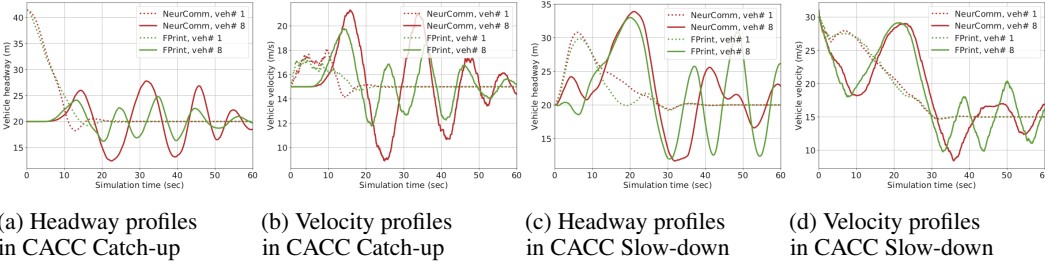

(a) Headway profiles in CACC Catch-up

(b) Velocity profiles in CACC Catch-up

(c) Headway profiles in CACC Slow-down

(d) Velocity profiles in CACC Slow-down

Figure 6: Headway and velocity profiles of the first and last vehicles of the platoon, controlled by top communicative and non-communicative policies in execution.

## 6    CONCLUSIONS

We have formulated the spatiotemporal MDP for decentralized NSC under neighborhood communication. Further, we have introduced the spatial discount factor to enhance non-communicative MARL algorithms, and proposed a neural communication protocol NeurComm to design adaptive and efficient communicative MARL algorithms. We hope this paper provides a rethink on developing scalable and robust MARL controllers for NSC, by following practical engineering assumptions and combining appropriate learning and communication methods rather than reusing existing MARL algorithms. One future direction is improving the recurrent units to naturally control spatiotemporal information flows within the meta-DNN in a decentralized way.

ACKNOWLEDGMENTS

We would like to thank Marco Pavone and Alexander Anemogiannis for valuable discussions and insightful comments.

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

## APPENDIX

## A PROOFS

### A.1 PROOF OF PROPOSITION 3.1

*Proof.* The proof follows the learning method in A2C Mnih et al. (2016), which shows that

$$\mathcal{L}(\theta) = \frac{1}{|\mathcal{B}|} \sum_{\tau \in \mathcal{B}} \left( -\log \pi_\theta(a_\tau|s_\tau)\hat{A}_\tau^\pi + \beta \sum_{a \in \mathcal{A}} \pi_\theta(a|s_\tau)\log \pi_\theta(a|s_\tau) \right), \tag{6}$$

$$\mathcal{L}(\omega) = \frac{1}{|\mathcal{B}|} \sum_{\tau \in \mathcal{B}} \left( \hat{R}_\tau^\pi - V_\omega(s_\tau) \right)^2, \tag{7}$$

where $\hat{A}_\tau^\pi = \hat{R}_\tau^\pi - v_\tau$, $\hat{R}_\tau^\pi = \sum_{\tau'=\tau}^{\tau_\mathcal{B}-1} \gamma^{\tau'-\tau}r_{\tau'} + \gamma^{\tau_\mathcal{B}-\tau}v_{\tau_\mathcal{B}}$, and $v_\tau = V_{\omega^-}(s_\tau)$, based on on-policy minibatch from a MDP $\{(s_\tau, a_\tau, r_\tau)\}_{\tau \in \mathcal{B}}$.

Now we consider spatiotemporal MDP, which has transition in Eq. (1), optimizes return in Eq. (2), and collects experience $(s_{i,t}, m_{\mathcal{N}_i i,t}, a_{i,t}, \tilde{r}_{i,t})$, where $\tilde{r}_{i,t} = \sum_{j \in \mathcal{V}} \alpha^{d_{ij}} r_{j,t}$. In Theorem 3.1 of Zhang

et al. (2018), the decentralized actor and critic are $\tilde{\pi}_{\theta_i}(s)$ and $\tilde{V}_{\omega_i}(s, a_{-i})$, for fitting $\pi_i^*(\cdot|s)$ and $\sum_{a_i \in \mathcal{A}_i} \pi_i(a_i|s)Q^{\pi_i}(s, a)$ under global observations, respectively. Now assuming the observations and communications are restricted to each neighborhood as in Definition 3.1, then the actor and critic become $\pi_{\theta_i}(\tilde{s}_i) \approx \tilde{\pi}_{\theta_i}(s)$ and $V_{\omega_i}(\tilde{s}_i, a_{\mathcal{N}_i}) \approx \tilde{V}_{\omega_i}(s, a_{-i})$, with the best observability.

Hence, replacing $\pi_\theta(a|s)$, $V_\omega(s)$, $r$ by $\pi_{\theta_i}(a_i|\tilde{s}_i)$, $V_{\omega_i}(\tilde{s}_i, a_{\mathcal{N}_i})$, and $\tilde{r}_i$, respectively, we establish Eq. (3)(4) from Eq. (6)(7), which concludes the proof. $\qquad\square$

Note partial observability and non-stationarity are present in $\pi_{\theta_i}(a_i|\tilde{s}_i)$ and $V_{\omega_i}(\tilde{s}_i, a_{\mathcal{N}_i})$. Fortunately, communication improves the observability. Based on Definition 3.1, any information that agent $j$ knows at time $t$ can be included in $m_{ji,t}$. We assume $s_{j,t} \cup \{m_{kj,t-1}\}_{k \in \mathcal{N}_j} \subset m_{ji,t}$. Then

$$
\begin{aligned}
\tilde{s}_{i,t} &\supset s_{i,t} \cup \left\{s_{j,t} \cup \{m_{kj,t-1}\}_{k \in \mathcal{N}_j}\right\}_{j \in \mathcal{N}_i} \\
&\supset \{s_{j,t}\}_{j \in \mathcal{V}_i} \cup \left\{s_{j,t-1} \cup \{m_{kj,t-2}\}_{k \in \mathcal{N}_j}\right\}_{j \in \mathcal{V}|d_{ij}=2} \\
&\supset \{s_{j,t}\}_{j \in \mathcal{V}_i} \cup \{s_{j,t-1}\}_{j \in \mathcal{V}|d_{ij}=2} \cup \left\{s_{j,t-2} \cup \{m_{kj,t-3}\}_{k \in \mathcal{N}_j}\right\}_{j \in \mathcal{V}|d_{ij}=3} \\
&\supset \dots \\
&\supset s_{i,t} \cup \left\{s_{j,t+1-d_{ij}}\right\}_{j \in \mathcal{V}\backslash\{i\}}.
\end{aligned}
$$

Thus, $\tilde{s}_{i,t}$ includes the *delayed* global observations. On the other hand, Eq. (1)(2) mitigate the non-stationarity. To see this mathematically,

$$
\begin{aligned}
\mathbb{E}_{\pi_i,p}[\tilde{r}_{i,t}|s_t, a_t] =& \mathbb{E}_{\pi_i,p_i}[r_{i,t}|s_{\mathcal{V}_i,t}, a_{\mathcal{N}_i,t}] + \alpha \sum_{j \in \mathcal{N}_i} \mathbb{E}_{\pi_i,p_j}[r_{j,t}|s_{\mathcal{V}_j,t}, a_{\mathcal{V}_j\backslash\{i\},t}] \\
&+ \sum_{d=2}^{d_{\max}}\left(\alpha^d \sum_{j \in \{\mathcal{V}|d_{ij}=d\}} \mathbb{E}_{p_j}[r_{j,t}|s_{\mathcal{V}_j,t}, a_{\mathcal{V}_j,t}]\right),
\end{aligned}
$$

where the further away reward signals are discounted more. Note if communication is allowed, each agent will have delayed global observations, and the non-stationarity mainly comes from limited information of future actions.

## A.2 PROOF OF PROPOSITION 4.1

This proposition contains two statements regarding neural communication based global information sharing in forward and backward propagations. We establish each of them separately.

**Lemma A.1** (Spatial Information Propagation). *In NeurComm, the delayed global information is utilized to estimate each hidden state, that is,*

$$
h_{i,t} \supset s_{i,0:t} \cup \left\{s_{j,0:t+1-d_{ij}}, \pi_{j,0:t-d_{ij}}\right\}_{j \in \mathcal{V}\backslash\{i\}}, \tag{8}
$$

*where $x \supset y$ if information $y$ is utilized to estimate $x$, and $x_{0:t} := \{x_0, x_1, \dots, x_t\}$.*

*Proof.* Based on the definition of NeurComm protocol (Eq. (5)), $m_{i,t} \supset h_{i,t-1}$, and $h_{i,t} \supset h_{i,t-1} \cup s_{\mathcal{V}_i,t} \cup \pi_{\mathcal{N}_i,t-1} \cup m_{\mathcal{N}_i,t}$. Hence,

$$
\begin{aligned}
h_{i,t} &\supset s_{i,t} \cup \{s_{j,t}, \pi_{j,t-1}\}_{j \in \mathcal{N}_i} \cup \{h_{j,t-1}\}_{j \in \mathcal{V}_i} \\
&\supset s_{i,t} \cup \{s_{j,t}, \pi_{j,t-1}\}_{j \in \mathcal{N}_i} \cup \left\{s_{j,t-1} \cup \{s_{k,t-1}, \pi_{k,t-2}\}_{k \in \mathcal{N}_j} \cup \{h_{k,t-2}\}_{k \in \mathcal{V}_j}\right\}_{j \in \mathcal{V}_i} \\
&= s_{i,t-1:t} \cup \{s_{j,t-1:t}, \pi_{j,t-2:t-1}\}_{j \in \mathcal{N}_i} \cup \{s_{j,t-1}, \pi_{j,t-2}\}_{j \in \{\mathcal{V}|d_{ij}=2\}} \\
&\qquad \cup \{h_{j,t-2}\}_{j \in \{\mathcal{V}|d_{ij}\leq 2\}} \\
&\supset \dots \\
&\supset s_{i,0:t} \cup \{s_{j,0:t}, \pi_{j,t-2:t-1}\}_{j \in \mathcal{N}_i} \cup \{s_{j,0:t-1}, \pi_{j,0:t-2}\}_{j \in \{\mathcal{V}|d_{ij}=2\}} \\
&\qquad \cup \dots \cup \{s_{j,0:t+1-d_{\max}}, \pi_{j,0:t-d_{\max}}\}_{j \in \{\mathcal{V}|d_{ij}=d_{\max}\}},
\end{aligned}
$$

which concludes the proof. $\qquad\square$

**Lemma A.2** (Spatial Gradient Propagation). *In NeurComm, each message is learned to optimize the performance of other agents, that is, $\{\nu_i, \lambda_i\}$ receive almost all gradients from $\mathcal{L}(\theta_j)$, $\mathcal{L}(\omega_j)$, $\forall j \in \{\mathcal{V}|j \neq i\}$.*

*Proof.* If we rewrite the required information for a given hidden state $h_{i,t}$ using intermediate messages instead of inputs, the result of Lemma A.1 becomes

$$
\begin{aligned}
h_{i,t} \supset \{m_{j,t}\}_{j \in \mathcal{N}_i} &\supset \{h_{j,t-1}\}_{j \in \mathcal{N}_i} \\
&\supset \{m_{j,t-1}\}_{j \in \{\mathcal{V}|d_{ij}=2\}} \supset \cdots \\
&\supset \{m_{j,t+1-d}\}_{j \in \{\mathcal{V}|d_{ij}=d\}} \supset \cdots
\end{aligned}
$$

Hence, $m_{i,\tau}$ is included in the meta-DNN of agent $j$ at time $\tau + d_{ij} - 1$. In other words, $\{\nu_i, \lambda_i\}$ receive gradients from $\mathcal{L}(\theta_j), \mathcal{L}(\omega_j), \forall j \in \{\mathcal{V}|j \neq i\}$, except for the first $d_{ij} - 1$ experience samples. Assuming $d_{\max} \ll |\mathcal{B}|$, $\{\nu_i, \lambda_i\}$ receive almost all gradients from loss signals of all other agents, which concludes the proof. $\square$

# B  ALGORITHMS

Algo. 1 presents the algorithm of model training in a synchronous way, following descriptions in Section 3 and 4. Four iterations are performed at each step: the first iteration (lines 3-5) updates and sends messages; the second iteration (lines 6-10) updates hidden state, policy, and action; the third iteration (lines 11-14) updates value estimation and executes action; the fourth iteration (lines 22-26) performs gradient updates on actor, critic, and neural communication. On the other hand, Algo. 2 presents the algorithm of decentralized model execution in an asynchronous way. It runs as a job that repeatedly measures traffic, sends message, receives messages, and performs control.

---

**Algorithm 1:** Multi-agent A2C with NeurComm (Training)

---

**Parameter :** $\alpha$, $\beta$, $\gamma$, $T$, $|\mathcal{B}|$, $\eta_\omega$, $\eta_\theta$.
**Result:** $\{\lambda_i, \nu_i, \omega_i, \theta_i\}_{i \in \mathcal{V}}$.

1  **initialize** $s_0$, $\pi_{-1}$, $h_{-1}$, $t \leftarrow 0$, $k \leftarrow 0$, $\mathcal{B} \leftarrow \emptyset$;
2  **repeat**
3      **for** $i \in \mathcal{V}$ **do**
4          **send** $m_{i,t} = f_{\lambda_i}(h_{i,t-1})$;
5      **end**
6      **for** $i \in \mathcal{V}$ **do**
7          **observe** $\tilde{s}_{i,t} = s_{\mathcal{V}_i,t} \cup \pi_{\mathcal{N}_i,t-1} \cup m_{\mathcal{N}_i,t}$;
8          **update** $h_{i,t} \leftarrow g_{\nu_i}(h_{i,t-1}, \tilde{s}_{i,t})$, $\pi_{i,t} \leftarrow \pi_{\theta_i}(\cdot|h_{i,t})$;
9          **update** $a_{i,t} \sim \pi_{i,t}$;
10     **end**
11     **for** $i \in \mathcal{V}$ **do**
12         **update** $v_{i,t} \leftarrow V_{\omega_i}(h_{i,t}, a_{\mathcal{N}_i,t})$;
13         **execute** $a_{i,t}$;
14     **end**
15     **simulate** $\{s_{i,t+1}, r_{i,t}\}_{i \in \mathcal{V}}$;
16     **update** $\mathcal{B} \leftarrow \mathcal{B} \cup \{(s_{i,t}, \pi_{i,t-1}, a_{i,t}, r_{i,t}, v_{i,t})\}_{i \in \mathcal{V}}$;
17     **update** $t \leftarrow t + 1$, $k \leftarrow k + 1$;
18     **if** $t = T$ **then**
19         **initialize** $s_0$, $\pi_{-1}$, $h_{-1}$, $t \leftarrow 0$;
20     **end**
21     **if** $k = |\mathcal{B}|$ **then**
22         **for** $i \in \mathcal{V}$ **do**
23             **update** $\hat{R}_\tau^{\pi_i}$, $\hat{A}_\tau^{\pi_i}$, $\forall \tau \in \mathcal{B}$, based on Proposition 3.1;
24             **update** $\{\lambda_j, \nu_j\}_{j \in \mathcal{V}} \cup \{\omega_i\}$, based on $\eta_w \nabla \mathcal{L}(w_i)$;
25             **update** $\{\lambda_j, \nu_j\}_{j \in \mathcal{V}} \cup \{\theta_i\}$, based on $\eta_\theta \nabla \mathcal{L}(\theta_i)$;
26         **end**
27         **initialize** $\mathcal{B} \leftarrow \emptyset$, $k \leftarrow 0$;
28     **end**
29 **until** *Stop condition is reached*;

---

---

**Algorithm 2:** Multi-agent A2C with NeurComm (Execution)

---

**Parameter :** $\{\lambda_i, \nu_i, \omega_i, \theta_i\}_{i \in \mathcal{V}}, \Delta t_{comm}, \Delta t_{control}$.

1 **for** $i \in \mathcal{V}$ **do**
2     **initialize** $h_i \leftarrow 0, \pi_i \leftarrow 0, \{s_j, \pi_j, m_j\}_{j \in \mathcal{N}_i} \leftarrow 0$;
3     **repeat**
4         **observe** $s_i$;
5         **update** $m_i \leftarrow f_{\lambda_i}(h_i)$;
6         **send** $s_i, \pi_i, m_i$;
7         **for** $j \in \mathcal{N}$ **do**
8             **receive** and **update** $s_j, \pi_j, m_j$ within $\Delta t_{comm}$;
9         **end**
10        **update** $\tilde{s}_i \leftarrow s_{\mathcal{V}_i} \cup \pi_{\mathcal{N}_i} \cup m_{\mathcal{N}_i}$;
11        **update** $h_i \leftarrow g_{\nu_i}(h_i, \tilde{s}_i), \pi_i \leftarrow \pi_{\theta_i}(\cdot | h_i)$;
12        **execute** $a_i \sim \pi_i$;
13        **sleep** $\Delta t_{control}$;
14     **until** *Stop condition is reached*;
15 **end**

---

## C   Experiment Details

### C.1   Algorithm Setup

Detailed algorithm implementations are listed below, in term of Eq. (5). IA2C: $h_{i,t} = \text{LSTM}(h_{i,t-1}, \text{relu}(s_{\mathcal{V}_i,t}))$. ConseNet: same as IA2C but with consensus critic update. FPrint: $h_{i,t} = \text{LSTM}(h_{i,t-1}, \text{concat}(\text{relu}(s_{\mathcal{V}_i,t}), \text{relu}(\pi_{\mathcal{N}_i,t-1})))$. NeurComm: $h_{i,t} = \text{LSTM}(h_{i,t-1}, \text{concat}(\text{relu}(s_{\mathcal{V}_i,t}), \text{relu}(\pi_{\mathcal{N}_i,t-1}), \text{relu}(h_{\mathcal{N}_i,t-1})))$. DIAL: $h_{i,t} = \text{LSTM}(h_{i,t-1}, \text{relu}(s_{\mathcal{V}_i,t}) + \text{relu}(\text{relu}(h_{i,t-1})) + \text{onehot}(a_{i,t-1}))$. CommNet: $h_{i,t} = \text{LSTM}(h_{i,t-1}, \tanh(s_{\mathcal{V}_i,t}) + \text{linear}(\text{mean}(h_{\mathcal{N}_i,t-1})))$. For ConseNet, we only do consensus update on the LSTM layer, since the input and output layer sizes may not be fixed across agents. Also, the actor and critic are $\pi_{i,t} = \text{softmax}(h_{i,t})$, and $v_{i,t} = \text{linear}(\text{concat}(h_{i,t}, \text{onehot}(a_{\mathcal{N}_i,t})))$

### C.2   Experiments in ATSC Environment

#### C.2.1   Action Space

Fig. 7 illustrates the action space of five phases for each intersection in the ATSC Grid scenario. The ATSC Monaco scenario has complex and heterogeneous action spaces, please see the code for more details. To summarize, there are 11 two-phase intersections, 3 three-phase intersections, 10 four-phase intersections, 1 five-phase intersection, and 3 six-phase intersections.

#### C.2.2   Summary of Execution Performance

Table 3 summarizes the key metrics in ATSC. The spatial average is taken at each second, and then the temporal average is calculated for all metrics (except for trip delay, which is directly aggregated over all trips). NeurComm outperforms all baselines on minimizing queue length and intersection delay. Interestingly, even though IA2C is good at optimizing the given objective of queue length, it performs poorly on optimizing intersection and trip delays.

#### C.2.3   Visualization of Execution Performance

Fig. 8 and Fig. 9 show screenshots of traffic distributions in the grid at different simulation steps for each MARL controller. The visualization is based on one execution episode with random seed 2000. Clearly, communicative MARL controllers have better performance on reducing the intersection delay. NeurComm and CommNet have the best overall performance.

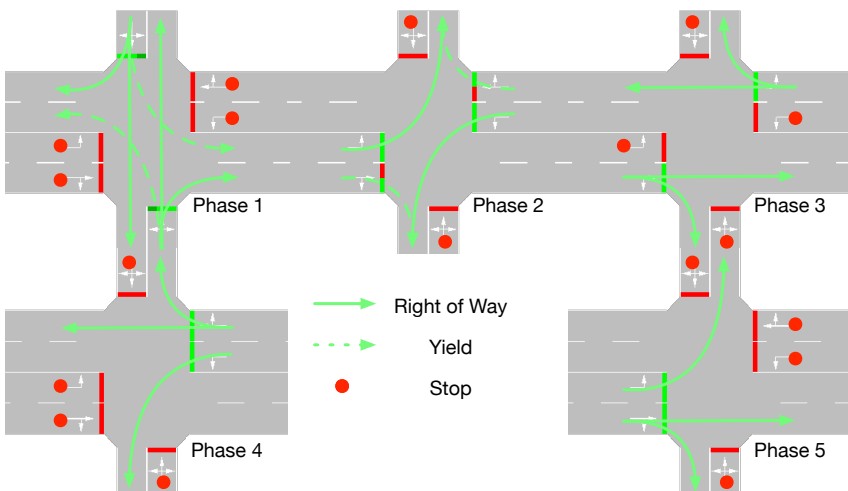

Figure 7: Possible signal phases at each intersection.

Table 3: Performance of MARL controllers in ATSC environments: synthetic traffic grid (top) and Monaco traffic network (bottom). Best values are in bold.

| Temporal Average Metrics | NeurComm | CommNet | DIAL | IA2C | FPrint | ConseNet |
|---|---|---|---|---|---|---|
| avg queue length [veh] | **1.16** | 1.44 | 2.36 | 1.63 | 1.62 | 2.04 |
| avg intersection delay [s/veh] | **68** | 111 | 145 | 376 | 415 | 366 |
| avg vehicle speed [m/s] | **2.28** | 1.82 | 1.78 | 0.26 | 0.23 | 0.29 |
| trip delay [s] | **293** | 455 | 1949 | 2067 | 1949 | 321 |
| avg queue length [veh] | **1.27** | 1.56 | 1.88 | 1.93 | 1.87 | 2.74 |
| avg intersection delay [s/veh] | 221.1 | 236.5 | 231.3 | **147.4** | 174.8 | 187.3 |
| avg vehicle speed [m/s] | 0.55 | 0.61 | 0.94 | **2.36** | 1.26 | 1.03 |
| trip delay [s] | 569 | 847 | 506 | **295** | 428 | 540 |

## C.3    EXPERIMENTS IN CACC ENVIRONMENTS

### C.3.1    SUMMARY OF EXECUTION PERFORMANCE

Table 4 summarizes the key metrics in CACC. The best headway and velocity averages are closest ones to $h^* = 20$m, and $v^* = 15$m/s. Note the averages are only computed from safe execution episodes, and we use another metric "collision number" to count the number of episodes where an collision happens within the horizon. Ideally, "collision-free" is the top priority. However, safe RL is not the focus of this paper so trained MARL controllers cannot achieve this goal in the experiments of CACC.

Table 4: Performance of MARL controllers in CACC environments: catch-up (above) and slow-down (below). Best values are in bold.

| Temporal Average Metrics | NeurComm | CommNet | DIAL | IA2C | FPrint | ConseNet |
|---|---|---|---|---|---|---|
| avg vehicle headway [m] | **20.45** | 20.47 | 21.99 | 22.02 | **20.44** | 21.45 |
| std vehicle headway [m] | 1.20 | 1.18 | 0.20 | 0.19 | 1.03 | 0 |
| avg vehicle velocity [m/s] | 15.33 | 15.33 | 15.07 | 15.07 | 15.33 | **15.00** |
| std vehicle velocity [m/s] | 0.90 | 0.87 | 0.16 | 0.18 | 0.75 | 0 |
| collision number | 0 | 0 | 0 | 0 | 0 | 0 |
| avg vehicle headway [m] | 15.84 | 16.24 | 14.42 | - | **18.21** | 11.60 |
| std vehicle headway [m] | 2.10 | 2.16 | 1.70 | - | 2.40 | 0.49 |
| avg vehicle velocity [m/s] | 13.43 | 13.82 | 12.28 | - | **15.47** | 8.59 |
| std vehicle velocity [m/s] | 2.77 | 2.88 | 2.49 | - | 3.37 | 1.19 |
| collision number | 13 | 12 | 16 | 50 | **8** | 23 |

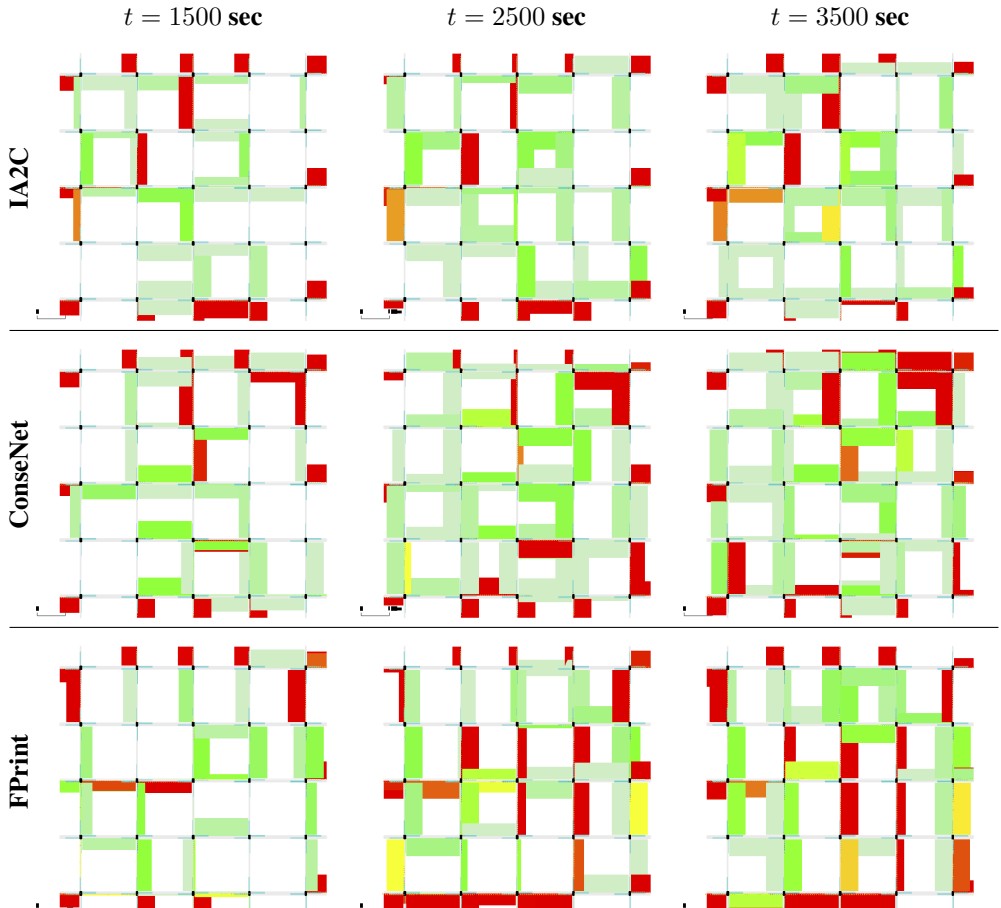

Figure 8: Screenshots of traffic distribution in the grid. Each row is a non-communicative MARL controller and each column is a simulation step. The traffic condition along each lane is visualized as a line segment, with the color indicating the queue length or congestion level (grey: 0% traffic, green: 25% traffic, yellow: 50% traffic, orange: 70% traffic, red: 90% traffic, intermediate traffic condition is shown as the interpolated color), while the thickness indicating the intersection delay (the thicker the longer waiting time at intersection).

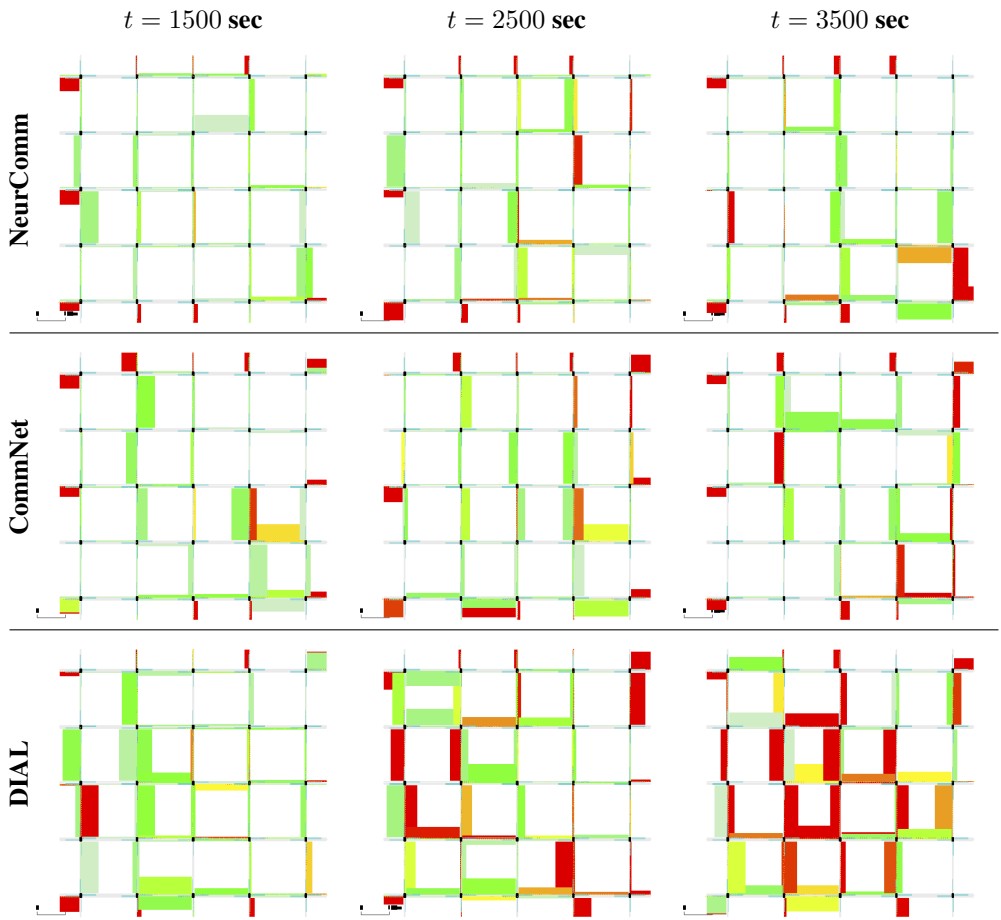

Figure 9: Screenshots of traffic distribution in the grid. Each row is a communicative MARL controller and each column is a simulation step.

