# OpenReview forum: "Multi-agent Reinforcement Learning for Networked System Control"
_ICLR.cc/2020/Conference — Accept (Poster)_

### Official Review · AnonReviewer2 · 2019-10-22
**Official Blind Review #2**

**Rating:** 6

**Review:**

This paper is concerned with network multi-agent RL (N-MARL), where agents need to update their policy based on messages obtained only from neighboring nodes. This is done under sensible restrictions on the state transition distribution, which can be claimed to hold true in realistic networked settings. The authors argue that introducing a spatial discount factor (along a temporal one), where neighboring nodes have a small distance, stabilizes learning. Also, they provide a way of learning a networked communication protocol. Experiments are done on somewhat realistic simulations of traffic.

I am not an expert in this type of distributed MARL, and what the SotA there is. What is proposed here makes sense to me, even though it also does not look very surprising to me. Spatial discounting seems to make sense. The authors do not comment on the relationship between the d_{ij} (or how to choose them) with the neighborhood graph, I suppose that neighbors have short distances, while non-neighbors have not. The NeurComm protocol is a bit vague to me, and I cannot say how it relates to other proposals. I find it a bit strange (unless I misunderstood something) that complete state and policy information is transmitted to every neighbor, and on top some latent vector. This sounds expensive to me, surely these communication channels are limited? It'd be more interesting to consider bandwidth limitations here. Without knowing related work on other protocols in detail, I suspect they probably work less well than NeurComm, simply because they allow for smaller messages only (f.ex., whereas they talk about "policy fingerprints", they seem to submit the complete policy parameters to neighbors -- that is not just a fingerprint). Maybe I am wrong, but if so, the paper does not explain things properly (there is some encoding and decoding going on, but just to propagate the hidden states at each node).

To me as an outsider, this looks like interesting work with well-simulated experiments. Of course, these are simulated and allow for no real-world conclusions, but then much of RL work does neither. It is just I'd be hard pressed to say what is really surprising in this paper, and what could be learned from it.


**Experience Assessment:**

I do not know much about this area.

**Review Assessment: Checking Correctness Of Derivations And Theory:**

I did not assess the derivations or theory.

**Review Assessment: Checking Correctness Of Experiments:**

I assessed the sensibility of the experiments.

**Review Assessment: Thoroughness In Paper Reading:**

I read the paper at least twice and used my best judgement in assessing the paper.

---

> ### Author Response · Authors · 2019-11-07
> **Clarification on contributions**
>
> We thank the reviewer for providing careful cross-domain assessment and relevant comments. Below is our response to the major comments.
>
> "SotA of distributed MARL."
> Typical SotA were compared against in this work, including CommNet, DIAL (communicative), and MADDPG, ConseNet (non-communicative), under the same DNN and training settings.
>
> "This work does not look very surprising."
> Please note the spatiotemporal MDP formulation (and the associated spatial discount factor) is completely new for MARL, though it is a natural extension from traditional (temporal) MDP. NeurComm also contains new features on top of SotA protocols, based on the understanding of non-stationarity and information loss in MARL.
>
> "Clarification on d_{ij} selection."
> d_{ij} here is the distance between nodes i and j in the graph (i.e. number of spatiotemporal MDP steps between agents i and j) rather than the physical distance.
>
> "How NeurComm is related to existing protocols? they probably work less well than NeurComm, simply because they allow for smaller messages only."
> As stated in Section 2 and 4, NeurComm differs from existing protocols as it 1) encodes and concatenates messages instead of aggregating them and 2) includes fingerprint in message. We agree that the additional fingerprint info helps NeurComm perform better. However, it is indeed our contribution to demonstrate that these particular neural-encoded neighborhood fingerprints would enhance the actor performance. For example, the recent work of mean-field RL (MFRL) demonstrated that additional information of averaged neighborhood policies would enhance the Q-value estimation (Yang, 2018). However, in heterogenous NMARL (e.g. ATSC Monaco) the neighbors have various action spaces and the aggregation becomes infeasible. Thus, NeurComm can be considered as a more general and robust communication protocol, based on a similar idea.
>
> Further, compared to existing protocols, the only additional message of each agent in NeurComm is its fingerprint, whose size is small enough compared to the size of belief (latent vector). In our experiments, the max fingerprint size is 6 whereas the belief size is 64.
>
> Also, the fingerprint is the same as policy in this work since A2C is on-policy. In off-policy algorithms such as DDPG, Q-learning, it has to be represented in other formats since behavior policy is not directly available (see (Foerster et al., 2017)).
>
> "Communication is expensive in practice."
> This is a good point. The community addresses communication scalability mostly from communication channels rather than message contents. In general MARL, each agent sends message to all other agents, leading to expensive global communication. So a collection of works focus on the attention mechanism. For each agent, the attention model gives a prioritized list of agents to be communicated with, so appropriate "neighborhood" can be formed to meet certain bandwidth limit. See the last paragraph of Section 2 for more details. However, in our problem setting, the neighborhood is already defined and usually small enough. Thus there is no need to apply attention further to select a subset of neighbors.
>
> We agree that scalability of message content should also be considered in practice. In our experiment, each message (start+policy+belief) has <100 floats so it is still affordable. Additional compression can be performed if the message is big. For example, we can add additional message output layer on top of the belief layer to meet the limit of message size.
>
> Ref:
> Yang, Yaodong, et al. "Mean field multi-agent reinforcement learning." arXiv preprint arXiv:1802.05438 (2018).
> Jakob Foerster, et al. "Stabilising experience replay for deep multi-agent reinforcement learning." arXiv preprint arXiv:1702.08887 (2017).

---

> > ### Comment · AnonReviewer2 · 2019-11-14
> > **Reaction to author feedback**
> >
> > I read the author feedback.
> > I now see that at least in their experiment (in a simulation), message sizes are still rather small, and so in this example, there won't be a problem.
> >
> > I have no reason to change my vote. It is still not clear to me whether their protocol just works better because more information is transformed. I am just missing the trade-off here. What if I transformed even much more information? When comparing protocols, a metric should be used which depends on message size as well. Communication will always be a major cost factor in practice.

---

> > > ### Author Response · Authors · 2019-11-14
> > > **Thanks for the feedback**
> > >
> > > We thank the reviewer for reading our response. We agree with the reviewer that the trade-off on message size vs performance should be considered. We will explicitly compare the message sizes across different protocols in the paper. As stated in our response, including fingerprint would lead to <= 10% increase in message size (additional 6 vs 64).
> > >
> > > Please note that "more information" does not always mean "better performance", we need more "relevant and effective" information. Fingerprint is relevant here since the MDP transition becomes more  stationary if each agent knows the neighbors' behavior policies (Eq.(1)). As stated in our response, some recent works (e.g. mean-field RL, FPrint RL) had similar ideas but different usages. However, no metrics/comparisons on message size were proposed so far. We will think more about the appropriate metrics.
> > >
> > > Thanks again for the constructive feedback.

---

### Official Review · AnonReviewer1 · 2019-10-23
**Official Blind Review #1**

**Rating:** 6

**Review:**

1. Summary

The authors use decentralized MARL for networked system control. Each agent might control a traffic light (exp 1) or a car in traffic (exp 2). Some features of their approach are a spatial Markov assumption (only neighborhood matters), a spatial discount factor, and NeurComm: a general message passing scheme between agent policies. The authors compare their method with CommNet (averages messages before broadcast), DIAL (small-scale direct communication), etc.

2. Decision (accept or reject) with one or two key reasons for this choice.

Weak accept.

3. Supporting arguments

The experiments and analysis of the more general communication scheme are nice and the assumptions used make sense for the environments considered (spatial interactions and dynamics).
The 2 environments used are nice, but it would be nice to see non-traffic applications as well.
The methodological contributions make sense in the spatial domain, but it would be interesting to see how neighborhood-based assumptions can be used in non-spatial / non-Euclidean domains.

4. Additional feedback with the aim to improve the paper. Make it clear that these points are here to help, and not necessarily part of your decision assessment.

It would be nice to explain why / what communication strategy enables FPrint to do better than NeurComm.

5. Questions

**Experience Assessment:**

I have published one or two papers in this area.

**Review Assessment: Checking Correctness Of Derivations And Theory:**

I assessed the sensibility of the derivations and theory.

**Review Assessment: Checking Correctness Of Experiments:**

I assessed the sensibility of the experiments.

**Review Assessment: Thoroughness In Paper Reading:**

I made a quick assessment of this paper.

---

> ### Author Response · Authors · 2019-11-07
> **Explanation on methodology and experiment**
>
> We thank the reviewer for the insightful comments and valuable suggestions. We are glad that the reviewer agreed on our approaches in experiment design and analysis. Below is our response to the major comments.
>
> "it would be nice to see non-traffic applications as well."
> We agree that including applications from different domains would increase the impact of this work. The dilemma was that, we wanted to focus more on the algorithm side in this work, say the spatiotemporal MDP w/ spatial discount factor, and the new NeurComm protocol. So we did not want to make it too "practical" (the experiment part already occupied half of total pages). Further, these environments actually covered various networked MARL topologies, such as homogenous 2D network (ATSC Grid), heterogenous 2D network (ATSC Monaco), and homogeneous 1D network (CACC), providing certain degree of generalization. Please see our response to Reviewer #3 ("More test cases are needed.") for more details.
>
> "it would be interesting to see how neighborhood-based assumptions can be used in non-spatial domains."
> Good suggestion! This work focuses on networked system control where the network topology is usually well-defined and static so the neighborhood-based assumption is automatically valid. However, our formulation can be extended to general non-network MARL environments as well, by integrating the attention mechanism. The high-level idea here is to form (dynamic) virtual "neighborhood" based on the impact strengths of all other agents on the target agent, which can be done by an attention mechanism (ATOC, IC3, etc), given a target neighborhood size (or bandwidth limit). Then our methods can be applied to this dynamic virtual network.
>
> "It would be nice to explain why / what communication strategy enables FPrint to do better than NeurComm."
> First, NeurComm still outperforms SotA communication protocols so this performance difference mainly comes from the difference between communicative vs non-communicative algorithms (Note FPrint only takes neighborhood information and does not have message-passing feature). Second, all communicative algorithms in experiment perform delayed information sharing (e.g. belief-message from d-distance away was generated d-step ago). Thus, this result implies that in real-time and mission-critical tasks such as CACC, delayed messages would distract the agent and hurt its performance, despite the additional information they provide. Please see our response to Reviewer #3 ("NeurComm does not outperform FPrint baseline in CACC applications.") for more details.

---

### Official Review · AnonReviewer3 · 2019-10-24
**Official Blind Review #3**

**Rating:** 6

**Review:**

This paper proposes a multiagent learning algorithm for networked system control. The main contributions are 1) a spatial discount factor that can stablize the learning process, 2) a differentiable communication protocol NeurComm. The paper was evaluated thoroughly by comparing against recent MARL baselines on two realistic environments.

I voted for "Weak Accept" because this paper can have important real-world applications such as traffic control, autonomous driving and power grid. I really like its evaluations on the realistic environments. In addition, the paper is clearly written, the algorithm seems reasonable and the evaluations are comprehensive.

I did not give it a higher rating because the proposed algorithm only outperforms the baselines in two out of four test cases. It is not a very strong result. In this situation, I would hope that see a more detailed discussion why in CACC experiments, NeurComm does not perform as well. And I would also suggest the paper adding more test cases to show that the proposed algorithm indeed can dominate in most of the cases.

**Experience Assessment:**

I have read many papers in this area.

**Review Assessment: Checking Correctness Of Derivations And Theory:**

I assessed the sensibility of the derivations and theory.

**Review Assessment: Checking Correctness Of Experiments:**

I assessed the sensibility of the experiments.

**Review Assessment: Thoroughness In Paper Reading:**

I read the paper at least twice and used my best judgement in assessing the paper.

---

> ### Author Response · Authors · 2019-11-06
> **Clarification and explanation on the experiment results**
>
> We thank the reviewer for assessing our work carefully and providing valuable comments. We are glad that the reviewer found this work an important step for real-world MARL applications in the future. Below is our response to the major comments.
>
> "NeurComm does not outperform FPrint baseline in CACC scenarios."
> First of all, we have not claimed that a single algorithm (e.g. NeurComm) would win in all real-world networked MARL (NMARL) environments. Actually, we do not believe such a universal algorithm exists, due to the severe partial observability and various MDP properties across real-world NMARL environments. Also, most of the recent MARL algorithms were only tested in multiple scenarios of a single fully observable environment rather than multiple partially observable environments, leading to similar experimental results (and potential overfitting). For example, MADDPG, IC3 (CommNet), ConseNet, Mean-field RL, ATOC were evaluated in only 2D space environment with freely moving dots (scenarios: cooperative navigation, predator-prey, battle game etc); FPrint, (Zhang, 2019), (Carion, 2019) were evaluated in only Starcraft environment (scenarios: 5m_vs_5m, 3s_vs_4z, 6h_vs_8z, etc).
>
> Further, rather than proposing a single algorithm, our contributions are indeed two novel methods: 1) \alpha to stabilize MARL under partial observability and 2) NeurComm to improve messages under non-stationarity. The effectiveness of both methods are clearly demonstrated in ablation study and experiments. Specifically, NeurComm outperforms SotA communication protocols even if it does not outperform FPrint (which is also not the baseline, but the one trained w/ \alpha) in CACC.
>
> Therefore, the statement is more like "communicative algorithms do not outperform \alpha-stabilized non-communicative algorithms in CACC", rather than "NeurComm does not outperform FPrint baseline in CACC". Recalling the major difference between NeurComm and FPrint is the belief-message passing, this implies that in real-time and mission-critical tasks such as CACC, additional delayed messages would distract the agent and hurt its performance (as stated in the first paragraph under Section 5.5). But this is the comparison between two algorithm groups (communicative vs non-communicative) and cannot be used to judge the effectiveness of NeurComm protocol. Similarly, we can imagine Q-learning (off-policy) would outperform A2C (on-policy) in this case since the replay buffer is more powerful in exploring good policies with sparse collision penalties, but this is neither the controlled evaluation on our method. So we re-implemented all methods in A2C even though some of them (FPrint, DIAL etc) were originally proposed in the context of Q-learning.
>
> We also proposed a way to overcome delays in information sharing via multi-pass communication (see the description under Eq. (5)). This will improve the performance of communicative algorithms in CACC but also bring scalability/latency issues in practice.
>
> "More test cases are needed."
> Thanks for the suggestion, we are planning to implement other realistic NMARL environments. However, this requires engineering efforts and domain-specific knowledge, and heavy experiment part would make this paper unbalanced, diluting the algorithm contributions (even in its current version, we had to defer most implementation details of CACC to appendix). So we selected two typical and popular real-world applications to show how our methods would improve the robustness and efficiency of NMARL in general. We plan to provide in-depth and domain-specific analysis in separate future works.
>
> Ref:
> Zhang, Sai Qian, Qi Zhang, and Jieyu Lin. "Efficient Communication in Multi-Agent Reinforcement Learning via Variance Based Control." arXiv preprint arXiv:1909.02682 (2019).
> Carion, Nicolas, et al. "A Structured Prediction Approach for Generalization in Cooperative Multi-Agent Reinforcement Learning." arXiv preprint arXiv:1910.08809 (2019).

---

### Decision · Program_Chairs · 2019-12-19

**Decision:**

Accept (Poster)

**Comment:**

The paper focuses on multi-agent reinforcement learning applications in network systems control settings. A key consideration is the spatial layout of such systems, and the authors propose a problem formulation designed to leverage structural assumptions (e.g., locality). The authors derive a novel approach / communication protocol for these settings, and demonstrate strong performance and novel insights in realistic applications. Reviewers particularly commended the realistic applications explored here. Clarifying questions about the setting, experiments, and results were addressed in the rebuttal, and the resulting paper is judged to provide valuable novel insights.